# The Relationship between Career Calling and Resilience among Rural-Oriented Pre-Service Teachers: The Chain Mediating Role of Career Adaptability and Decent Work

**DOI:** 10.3390/bs14010011

**Published:** 2023-12-22

**Authors:** Ya Wen, Huaruo Chen, Fei Liu, Xindong Wei

**Affiliations:** 1School of Teacher Education, Nanjing XiaoZhuang University, Nanjing 211171, China; wenya@njxzc.edu.cn; 2College of Education Science and Technology, Nanjing University of Posts and Telecommunications, Nanjing 210023, China; 3Center for Research and Reform in Education, Johns Hopkins University, Baltimore, MD 21286, USA; 4School of Education Science, Nanjing Normal University, Nanjing 210046, China; 190601009@njnu.edu.cn; 5School of Teacher Education, Huaiyin Normal University, Huaian 223001, China; 6School of Teacher Education, Nanjing University of Information Science & Technology, Nanjing 210044, China; 003454@nuist.edu.cn

**Keywords:** Psychology of Working Theory, career calling, career adaptability, decent work, resilience, rural-oriented pre-service teachers

## Abstract

From a Psychology of Working Theory (PWT) perspective, this study aims to explore how career adaptability and decent work mediate the effects of career calling on the resilience of rural-oriented pre-service teachers. The Career Calling Scale, Career Adaptability Scale, Decent Work Scale, and Resilience Scale were used to survey 393 rural-oriented pre-service teachers. The results found that career calling positively predicted the resilience of rural-oriented pre-service teachers; career adaptability mediated the relationship between career calling and resilience of rural-oriented pre-service teachers; decent work mediated the relationship between career calling and resilience of rural oriented pre-service teachers; and career adaptability and decent work play a chain mediating role between career calling and career adaptability of rural-oriented pre-service teachers. Therefore, this study concludes that career calling not only directly influences the resilience of rural-oriented pre-service teachers, but also indirectly influences the resilience of rural-oriented pre-service teachers through career adaptability and decent work.

## 1. Introduction

Rural education, essential for the development of rural talents, relies heavily on a high-quality teaching force [1]. Teaching is a respected profession in China, and most teachers receive their training from universities at the pre-service stage, where they are nurtured to develop their talents [2]. Rural-oriented pre-service teachers, nurtured by Chinese universities, are individuals specifically trained by the government to address the current teaching resource shortage in rural areas, particularly in remote and struggling regions [3]. After completing their undergraduate studies, these college students return to their hometowns to teach in village schools, aiming to provide better educational opportunities for rural children and further educational equity in society. Considering the disparities between rural and urban educational resources and conditions, some studies have found that these rural-oriented pre-service teachers face mental health challenges that might adversely affect their career development, necessitating adjustments as they transition from university to their teaching roles in rural settings [4,5]. Resilience, integral to mental health topics such as depression and anxiety, refers to the vital qualities and abilities that enable individuals to thrive in adversity and stress, influenced by both their inner and outer circumstances [6,7,8]. During a teacher’s career development, resilience serves as a powerful and constructive psychological asset. Given the heightened teaching pressures, increased workloads, and potential for job burnout, it is crucial for pre-service teachers to cultivate resilience early in their careers [9]. Nevertheless, there is a dearth of current research related to the resilience of pre-service teachers [10]. For this reason, the factors that may influence the resilience of pre-service teachers have received the attention of teacher education researchers in a number of countries. A sense of career calling is a transcendent summons, experienced as originating outside of the self, guiding individuals to approach life roles with a profound sense of meaning centered on other-oriented values and goals as their primary motivation [11,12,13]. Career calling, once a significant research topic in the career field, has recently garnered attention in the realm of teacher education. For teachers, a sense of career calling usually leads to positive psychological outcomes: individuals with this calling typically exhibit greater career adaptability, reduced turnover intentions, and heightened well-being [14,15,16]. While a sense of career calling might predict an individual’s resilience, the specific underlying mechanisms remain unexplored.

The Psychology of Working Theory (PWT) is an emerging theory that has received widespread attention in the career field in recent years, aiming to interpret the work experiences of a diverse group of people using a theoretical model that involves several variables related to career development and mental health, such as career adaptability [17,18]. PWT is a highly inclusive career theory that attaches particular attention to the access to decent work of the poor, marginalized, or disadvantaged groups who face challenges in their lives [19,20]. At the heart of PWT, decent work encompasses safe working conditions, adequate rest time, fair pay, comprehensive health insurance, and work values that align with family values [21,22]. At present, from a PWT perspective, there is limited research on the factors influencing an individual’s mental health and positive psychological traits. Thus, this study endeavors to investigate how a sense of career calling and associated variables influence the resilience of rural-oriented pre-service teachers, using the PWT framework as a foundation.

## 2. Literature Review

### 2.1. Resilience

Resilience, an essential psychological variable closely associated with individual mental health, is recognized as a construct that can be cultivated [1,23]. Resilience is often regarded as a critical ability to use personal and situational resources to successfully cope with challenges in the face of adversity such as health, relationships, and financial problems, and is a core component of personal stress management [24]. Specifically, resilience is a developable and positive mental capacity that characterizes the process and outcome of an individual’s successful adaptation to difficult or challenging life experiences, particularly through mental, emotional and behavioral flexibility and adjustment to internal and external demands [25]. Resilience of an individual plays a pivotal role in achieving career success, influencing aspects like employment opportunities, workplace advancement, and overall future career development.

Previous research on resilience has mainly concentrated on the career development and emotions of individuals. For instance, studies have revealed that the resilience of college students is positively correlated with mindfulness, optimism and life satisfaction [26,27,28,29]. Conversely, individuals with lower resilience levels are more likely to face challenges in achieving success and well-being [30,31]. With regard to teachers, educators propose that fostering resilience among teachers is necessary in order to maintain or restore their well-being and commitment to quality education which contributes to their career development [23]. Studies have demonstrated that the resilience of teachers is strongly and positively related to well-being, and negatively related to job burnout and turnover intention [32]. Teachers with higher resilience levels can manage stress effectively, adapt to students of varied personalities, and maintain a positive outlook [33]. Rural-oriented pre-service teachers need to be well and psychologically resilient. While it is vital they complete their university studies successfully or become qualified rural teachers in the future, there are limited investigations into the resilience of this specific group.

### 2.2. Career Calling and Resilience

In the realm of teacher education, career calling is a pivotal concept that plays a vital role in individual career development, which is anchored in the belief that one’s career in teaching is not just a job but a central aspect of a broader sense of purpose [14,34]. It serves to assist students, foster learning, and promote the greater good in society. In the Chinese cultural context, in the context of teaching, ‘calling’ is perceived as a profound and honorable term. It signifies a career orientation steered by both an external summons and an intrinsic desire, rooted in personal beliefs and values, that emphasizes the significance of education for both the individual and society [12,35]. Career calling has been categorized by Chinese scholars into three dimensions: altruism, guiding force, and meaning and purpose [36]. Altruism is the willingness of individuals with a sense of mission to assist others, to serve the public interest, or even the well-being of society as a whole; guiding force refers to the guiding influence on career development, such as responsibility or a sense of duty; and meaning and purpose underscores the idea that educators, driven by a deep vocational commitment, derive profound significance and purpose from their chosen profession [36].

Teaching is an emotionally demanding profession, which makes a sense of career calling especially important for teachers, given the special nature of the profession. If teachers have a sense of career calling, they are more likely to recognize the value of their work, not just in terms of external material and monetary rewards, but also to satisfy their inner needs and realize their self-worth, so as to obtain the meaning of life [37]. Career calling usually leads to many positive consequences, for instance, it was discovered that the stronger the career calling, the higher the job satisfaction and life satisfaction of Korean teachers [38]. Moreover, a study focusing on Chinese teachers concluded that a sense of career calling is negatively related to burnout and positively related to work engagement, psychological capital, gratitude, and meaning in life [39,40]. Individuals with a higher sense of career calling are more apt to make adequate plans and preparations to meet challenges when they are faced with difficult situations in their careers, potentially leading to increased psychological resilience.

### 2.3. The Mediating Role of Career Adaptability

Career adaptability, an essential component of the PWT framework, has a beneficial influence on an individual’s well-being [19]. It is a crucial psychological resource that affects an individual’s career development [41]. Career adaptability allows individuals to handle both predictable tasks and unpredictable adjustments due to changes in work conditions, preparing them for various work roles [42]. In essence, it equips an individual to navigate career role changes smoothly [43]. Career adaptability encompasses four dimensions: concern, control, curiosity, and confidence [44]. Career concern is the tendency to plan and anticipate one’s career future; career control is the ability to take responsibility for one’s career; career curiosity is the exploration of one’s career role with an open mind to increase self-knowledge and career knowledge; and career confidence refers to the ability to pursue career success despite the challenges one faces [45]. For employees, higher career adaptability is associated with a lower intention to leave their job [45]. Research on college students indicates that higher career adaptability leads to increased career engagement, better academic satisfaction, improved job search outcomes, and a greater likelihood of adopting proactive career advancement behaviors [46,47].

Career adaptability is a crucial quality for the professional development of teachers. Research involving Chinese pre-service teachers has identified a positive correlation between career calling and career adaptability [14]. In studies using nursing undergraduates, scholars have observed a positive correlation between career adaptability and resilience [48]. Faculty-based studies have also indicated a positive correlation between career adaptability and factors like career optimism [49].

### 2.4. The Mediating Role of Decent Work

In recent years, the examination of decent work, grounded in the PWT, has gained increasing attention from researchers. Specifically, decent work, as an occupational outcome, signifies quality employment characterized by safe working conditions, adequate rest breaks, adequate remuneration, health-care coverage and the matching of work values with family values [22]. It represents the central variable within the PWT framework, and is commonly delineated into five elements: safe working conditions, adequate health care, suitable compensation, sufficient rest periods, and organizational values aligned with family and society [19]. All the variables in the PWT framework have been connected around the central variable of decent work. Based on the PWT, predictors of decent work include marginalization, economic constraints, career adaptability, and work volition [19]. The outcomes of decent work encompass need satisfaction and the acquisition of well-being [50]. For individuals, decent work can affect well-being both directly and indirectly through the satisfaction of basic human needs [51].

Based on the PWT, researchers have revealed that higher levels of decent work result in more need satisfaction, which leads to higher job satisfaction and well-being, and lower turnover intention for individuals [50,52]. For university students, who are typically not yet employed, the researchers detected a positive correlation between future decent work perceptions and the level of academic satisfaction and career exploration [53,54]. Put simply, in rural-oriented pre-service teachers at university level, individuals with a greater awareness of decent work tend to approach challenges in school or life with a more positive perspective and, as a result, are better equipped to handle career crises. Additionally, given the unique significance of a career calling in the teaching profession, a higher sense of career calling might be correlated with rural-oriented pre-service teachers’ future decent work perceptions.

### 2.5. The Chain Mediating Role of Career Adaptability and Decent Work

For individuals within the teaching profession, the sense of a career calling plays a pivotal role in influencing their career development. It has been identified that career calling positively correlated with career adaptability [14]. Career adaptability, a multidimensional psychosocial construct linked to an individual’s long-term career success, is the ability to handle career development-related tasks using available resources [55,56,57]. A survey based on Chinese pre-service female primary school teachers found that greater career adaptability was associated with higher individual perceptions of future decent work and higher academic satisfaction [53]. In summary, when career adaptability is high, individuals tend to remain curious about external workplace changes, more confident about their own career development thus forming more objective and clearer perceptions of future decent work, and are more likely to seek more suitable solutions with higher resilience when confronted with career difficulties. However, according to the PWT framework, scholars have not yet explored the impact of career calling, quality employment, and career adaptability on resilience, specifically among rural-oriented pre-service teachers.

### 2.6. Hypothesis

In order to investigate the relationship between career calling and resilience of rural-oriented pre-service teachers under the PWT model, the following 9 hypotheses were formulated:

**Hypothesis 1 (H1):** 
*career calling positively predicts resilience.*


**Hypothesis 2 (H2):** 
*career calling positively predicts career adaptability.*


**Hypothesis 3 (H3):** 
*career adaptability positively predicts resilience.*


**Hypothesis 4 (H4):** 
*career calling positively predicts decent work.*


**Hypothesis 5 (H5):** 
*career adaptability positively predicts decent work.*


**Hypothesis 6 (H6):** 
*decent work positively predicts resilience.*


**Hypothesis 7 (H7):** 
*career adaptability plays a partial mediating role in career calling and resilience.*


**Hypothesis 8 (H8):** 
*decent work plays a partial mediating role in career calling and resilience.*


**Hypothesis 9 (H9):** 
*career adaptability and decent work play the role of chain intermediary in the relationship between career calling and resilience.*


The hypothetical model is shown in Figure 1.

## 3. Materials and Methods

### 3.1. Participants

In this study, a total of 420 students from Nanjing Xiaozhuang University in eastern China participated in an online questionnaire. Nanjing Xiaozhuang University is committed to the cultivation of educators for primary and secondary schools. The university places a strong emphasis on teacher education, offering comprehensive programs to equip students with the knowledge and skills needed for successful teaching careers. During the questionnaire distribution process, we contacted college staff from the university to obtain their support for the survey. Before starting the questionnaire, all participants were informed about its purpose and assured of the confidentiality of their personal information. All participants voluntarily took part in the questionnaire. The questionnaire was returned to 393 participants (48 males) with a validity rate of 94%. Of the participants, 26% were in their first year of college, 25% were in their second year of college, 28% were in their third year of college, and 21% were in their fourth year of college. This study was conducted in May 2022.

### 3.2. Instruments

#### 3.2.1. Career Calling

The Career Calling Scale was formulated by Zhang [58,59]. The scale was composed of 3 related subscales: altruism, guiding force, and meaning and purpose, with a total of 11 items. The scale uses a 5-point Likert scale (1 = strongly disagree; 5 = strongly agree) in this study. A higher score for the participant indicates that the individual has a stronger sense of career calling. The Cronbach’s alpha coefficient of the scale was 0.91. Confirmatory factor analysis results showed that the three-factor model fit the data: χ^2^/df = 3.123, RMSEA = 0.074, NFI = 0.970, IFI = 0.980, TLI = 0.972, and CFI = 0.980. Basic information about the scale can be found in Appendix A, Table A1.

#### 3.2.2. Career Adaptability

The Career Adapt-Abilities Scale—Short Form (CAAS-SF) was developed by Yu et al. [60,61]. There were 12 items and 4 dimensions: concern, control, curiosity, and confidence [14]. A five-point Likert score was adopted (1 = strongly disagree; 5 = strongly agree). The higher a participant’s score, the better their career adaptability. In this study, the Cronbach’s alpha coefficient of the scale was 0.95. Confirmatory factor analysis results revealed that the four-factor model fit the data: χ^2^/df = 3.482, RMSEA = 0.080, NFI = 0.967, IFI = 0.976, TLI = 0.960, and CFI = 0.976. Basic information about the scale can be found in Appendix A, Table A2.

#### 3.2.3. Future Decent Work Perceptions

Future decent work perceptions were assessed with a scale revised by Wei et al., designed to examine students’ views on their future work [22,62]. The scale consists of 15 questions distributed across 5 subscales, namely workplace safety, health care, adequate compensation, free time and rest, and organizational values that match family and social values [63]. Responses were gathered using a 7-point Likert scale, ranging from 1 to 7 (1 = strongly disagree, 7 = strongly agree). In this research, the Cronbach’s alpha coefficient was found to be 0.98. Confirmatory factor analysis results indicated that the five-factor model fit the data: χ^2^/df = 3.331, RMSEA = 0.077, NFI = 0.973, IFI = 0.981, TLI = 0.973, and CFI = 0.981.Basic information about the scale can be found in Appendix A, Table A3.

#### 3.2.4. Resilience

Resilience was measured with a revised scale designed by Connor and Davidson [32,64]. The scale was made up of 25 items across 3 dimensions: confidence, optimism, and strength. A 5-point Likert scale was used (1 = very inconsistent; 5 = very consistent). In the present study, the Cronbach’s alpha coefficient was 0.96. Confirmatory factor analysis results demonstrated that the three-factor model fit the data: χ^2^/df = 3.107, RMSEA = 0.073, NFI = 0.902, IFI = 0.931, TLI = 0.917, and CFI = 0.931. Basic information about the scale can be found in Appendix A, Table A4.

### 3.3. Data Analysis

To ascertain that the scales had acceptable psychometric properties, SPSS 25.0 was implemented to analyze the data, examining common method bias, the normality of the data, and assessing the reliability of the scales through Cronbach’s alpha coefficients. To confirm the appropriateness of the sample distribution for the subsequent analyses, descriptive statistics were employed, and correlations among the four variables were assessed to determine the feasibility of building the model. Using Amos 24.0 Structural Equation Modeling, a comprehensive model was constructed to examine the relationship between rural-oriented pre-service teachers’ career calling, career adaptability, decent work, and resilience.

## 4. Results

### 4.1. Common Method Deviation Test and the Normality of the Data

Four scales were used in this study, all of which underwent consistent testing. Harman’s one-factor test was applied to check for common method bias. Factors such as the content of the scales, the testing environment, and participant characteristics might introduce bias in the findings [65]. The first unrotated factor, retrieved from the factor analysis and incorporating all items, was close to 40%. The skewness (−0.98~0.28) and kurtosis (−0.13~2.25) of all variables entering the SEM were under the recommended limits [66].

### 4.2. Correlations among Variables

Correlation analysis, as presented in Table 1, revealed that career calling was significantly positively correlated with resilience (r = 0.53, *p* < 0.01). Hypothesis 1 was supported. Career calling was significantly positively correlated with career adaptability (r = 0.60, *p* < 0.01). Hypothesis 2 was supported. Career adaptability was significantly positively correlated with resilience (r = 0.68, *p* < 0.01). Hypothesis 3 was supported. Career calling was significantly positively correlated with future decent work perceptions (r = 0.52, *p* < 0.01). Hypothesis 4 was supported. Career adaptability was significantly positively correlated with future decent work perceptions (r = 0.63, *p* < 0.01). Hypothesis 5 was supported. Future decent work perceptions were significantly positively correlated with resilience (r = 0.72, *p* < 0.01). Hypothesis 6 was supported.

### 4.3. Mediating Effect and Final Structural Model

Immediately after, we generated a bootstrap 95% confidence interval to assess the potential mediating effects. Specifically, the significance of the mediating effect was further tested in this study using the bias-corrected bootstrap method with 5000 repetitions. We found that the total effect of career calling on resilience was significant (Effect = 0.487, SE = 0.150, 95% CI [1.526, 2.116]) and the direct effect of career calling on resilience was significant (Effect = 0.084, SE = 0.037, 95% CI [0.010, 0.157]). Previous literature has proven that the indirect effect can be substantiated if the confidence interval does not contain zero [67]. The results indicated that career adaptability had a mediating effect between career calling and resilience. The standardized effect value of Path 1 was 0.184, with a 95% confidence interval [0.109, 0.294], indicating that the mediating effect was significant. Hypothesis 7 was verified. Path 2 had a standardized effect value of 0.095 with a 95% confidence interval of (0.030, 0.160), indicating that the mediating impact was significant. The standardized effect value of the chain mediating effect of career adaptability and future decent work perceptions between career calling and resilience was 0.124. The 95% confidence interval (0.079, 0.182) did not include 0, demonstrating the significance of the chain mediating effect. Hypothesis 8 was verified. The results above suggested that career adaptability and future decent work perceptions play a chain mediating role between career calling and resilience. Thus, Hypothesis 9 was verified. Refer to Table 2 for detailed statistical outcomes, which were tested by PROCESS.

The present study developed a structural equation model of career calling and resilience to further validate the chain mediating role of career adaptability and future decent work perceptions. Previous studies suggest that the fitted value of a good model needs to reach the following criteria, for example, χ^2^/df values below 3 and an RMSEA values below 0.08 [68]. In this investigation, the fit indices of the models were as follows χ^2^/df = 3.995, RMSEA = 0.087, TLI = 0.946, CFI = 0.958, NFI = 0.945, RFI = 0.929, and IFI = 0.958. Although the χ^2^/df and RMSEA values are slightly above the recommended thresholds, the other indices suggest that the model has a satisfactory fit. In view of this, we concluded that the model as presented in the previous stage has a good fit with the actual data. This suggests that career adaptability and future decent work perceptions serve as chain mediators between career calling and resilience. The structural equation model developed in this research by Amos was presented in Figure 2.

## 5. Discussion

The main purpose of this research was to explore the influence of career calling on the resilience of rural-oriented pre-service teachers within the framework of PWT, and the mediating role of career adaptability and future decent work perceptions in this relationship. Firstly, it has been revealed that resilience can be positively predicted by career calling. For rural-oriented pre-service teachers, career calling consists of the experience of regarding the teaching profession as a life goal. It refers to the strong enthusiasm that individuals experience when working in a specific field of education, which plays an influential role in the career development of individuals at the pre-service teacher stage and provides a positive drive [14]. Rural-oriented pre-service teachers with a heightened sense of career calling are not only more inclined to accept the social role requirements of teachers and appreciate the intrinsic value of education work, but also engage more actively in education and teaching with a stronger sense of work responsibility. During this journey, they might face various academic or personal challenges. Yet, their profound interest and passion for the teaching profession empower them to maintain a positive outlook, making them more resilient in the face of adversity.

Secondly, this study found that among rural-oriented pre-service teachers, career adaptability acted as a mediator between career calling and resilience. Previous research focusing on pre-service teachers found that career calling positively correlates with career adaptability [14]. In addition, past studies have established a strong relationship between elevated levels of career adaptability, and both optimism and well-being [49,69]. Consistent with prior studies, this research indicates that among rural-oriented pre-service teachers, a higher degree of career calling correlates with improved career adaptability. This, in turn, enhances their resilience and ability to recover from potential career challenges.

Thirdly, decent work was observed to act as a mediator between career calling and resilience of rural-oriented pre-service teachers. Another study explored the relationship between decent work and both physical and mental health. It found that decent work not only promotes mental health but also supports physical health [70]. In addition, a survey based on female workers revealed a positive association between decent work, well-being and work fulfillment [71]. Similar to previous studies, individuals with a stronger sense of career calling often experience higher levels of decent work. This, in turn, leads to positive psychological utility and assists individuals to have higher resilience in the face of adversity.

Finally, our survey provides evidence that career adaptability and future decent work perceptions act as chain mediators between career calling and resilience. An individual with a strong sense of career calling may develop greater career adaptability, leading to a positive perception of future work, which in turn could bolster their resilience. Rural-oriented pre-service teachers with a strong career calling can not only appreciate the significance and value of teaching but are also inclined to have enhanced career adaptability—evident in their career concern, curiosity, control, and confidence. Such teachers are likely to have a positive outlook on future work, understand challenges in remote rural settings, and thus, display greater resilience [49].

## 6. Implications in Practice

The outcome of this study offers several practical implications for both practitioners and administrators. To better prepare rural-oriented pre-service teachers for a prosperous career, we need to bring the following practical recommendations to the attention of the relevant authorities.

The results of this investigation demonstrate that future educational practice may require university-level development of career calling among rural-oriented pre-service teachers. First, teacher education institutions at universities responsible for preparing rural-oriented pre-service teachers need to emphasize the development of a sense of career calling among teacher educators across various professional curricula. By integrating the three dimensions of career calling—altruism, guiding force, and meaning and purpose—into the teacher education curriculum, we can effectively convey the state and society’s expectations for teachers to rural-oriented pre-service educators, thereby fostering a strong and positive professional identity among them [32,72,73]. Second, universities offering career education to rural-oriented pre-service teachers should instruct them to think seriously about the characteristics of the teaching profession, encourage them to consider teaching as their career ideal, and cultivate a strong belief in lifelong teaching.

Simultaneously, education departments should be sensitive to the fundamental importance of fostering correct work perceptions for the career development of rural-oriented pre-service teachers. On the one hand, the government takes the need to develop educational responses to support students at this stage to form a clear and positive perception of work, and to help rural-oriented pre-service teachers consciously identify with the careers they will pursue and actively assume the corresponding professional roles and social expectations. On the other hand, the education department offers various supports to enhance the career adaptability of rural-oriented pre-service teachers. Throughout teacher training, students are offered various supports to assist them in focusing on their own careers, enhancing their career confidence, and striving to improve their professional abilities, so as to form a correct concept of work [74].

In conclusion, resilience has many positive implications for the career development of rural-oriented pre-service teachers. Therefore, to enhance the resilience of this group, career counselors and other educators in universities should design appropriate intervention programs. Specifically, during their university education instructors and counselors for rural-oriented pre-service teachers should emphasize career calling, career adaptability, and future decent work perceptions as key elements in their courses and activities. This will enable them to create and participate in teaching skills competitions and carry out various kinds of scientific research projects related to teacher education. These efforts can further bolster student resilience and better prepare them for their future teaching career.

## 7. Limitations and Future Directions

This work has several limitations that future researchers should address. First, there may be a single-method bias issue in this study. In this study, all data were sourced from participants’ self-reports [75]. Second, this study utilized a cross-sectional design, which implies that all data in the study were gathered at the same point in time, making it challenging to establish a causal relationship between the research variables [21,76]. Third, other mediating mechanisms, such as work volition, might influence career calling on resilience [77]. Fourth, the sample was selected from rural-oriented pre-service teachers, a group representing only one type of pre-service teachers and therefore not encompassing the diversity of this profession [53]. Despite these limitations, this research supports the enrichment of career studies of rural-oriented pre-service teachers and provides a foundation for more rigorous research designs in the future.

In general, there are likely to be several possible directions for future researchers. First, future researchers could engage in more diverse research methods, such as qualitative research or mixed research [78]. For instance, the relationship between career calling, career adaptability, future decent work perceptions and resilience from a PWT can be more thoroughly and comprehensively interpreted by researchers through the form of in-depth interviews [79]. Also, longitudinal examinations or more complex experimental designs may be employed by researchers to facilitate causal inferences about the development of resilience in this group based on a PWT perspective [80]. In addition, subsequent researchers should consider deepening the research design, especially by improving the mediating mechanism, to validate the research conclusions more effectively. Researchers can further explore mediating variables between career calling and resilience, aiming to propose more comprehensive interventions for rural-oriented pre-service teachers, drawing from a broader range of variables. Finally, it may be beneficial for researchers to incorporate larger and more diverse participant groups into the PWT theoretical framework to, further enriching and validating the theory.

## 8. Conclusions

In conclusion, the current study is a validation and expansion of the relationship between variables under the PWT, thus enriching the field of the career development of rural-oriented pre-service teachers. The following main results were attained in this study: career calling significantly predicted resilience in rural-oriented pre-service teachers; and career adaptability and future decent work perceptions partially mediated the relationship between career calling and resilience. Despite the limitations, this study reveals the relationship between career calling, resilience, future decent work perceptions, and resilience of rural-oriented pre-service teachers in China from the perspective of PWT, which has unique and significant practical implications relevant to those working in teacher education and to professional counselors.

## Figures and Tables

**Figure 1 behavsci-14-00011-f001:**
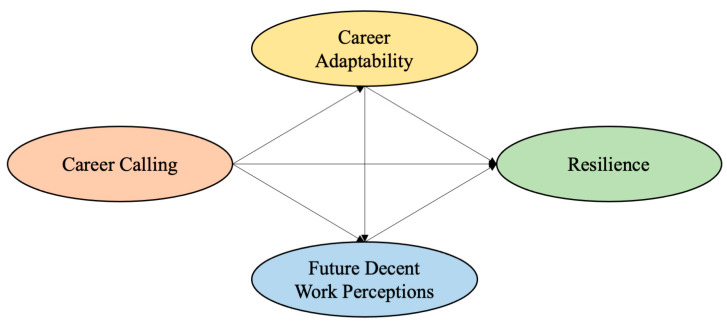
Hypothesis Model.

**Figure 2 behavsci-14-00011-f002:**
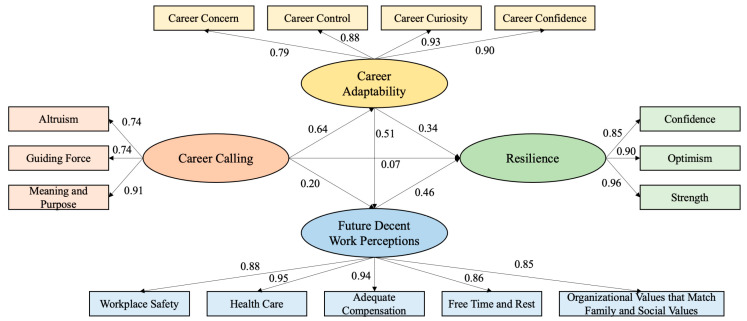
Standardized Relationship Path Diagram.

**Table 1 behavsci-14-00011-t001:** Descriptive statistics and correlations of all variables (*n* = 393).

Variable	*M*	*SD*	1	2	3	4
1 Career calling	3.91	0.66	-			
2 Career adaptability	3.99	0.62	0.60 **	-		
3 Future decent work perceptions	5.35	1.04	0.52 **	0.63 **	-	
4 Resilience	3.66	0.61	0.53 **	0.68 **	0.72 **	-

Note: ** *p* < 0.01.

**Table 2 behavsci-14-00011-t002:** Bootstrap analysis of mediating effect significance test.

	Effect Value	Boot SE	95% Confidence Interval		*p*
			Boot LLCI	Boot ULCI	
Total effect	0.487	0.150	1.526	2.116	0.000
Direct effect	0.084	0.037	0.010	0.157	0.026
Total indirect effect	0.403	0.057	0.306	0.527	0.000
Path 1	0.184	0.047	0.109	0.294	0.026
Path 2	0.095	0.033	0.030	0.160	0.000
Path 3	0.124	0.026	0.079	0.182	0.000

Path 1: career calling—career adaptability—resilience; Path 2: career calling—future decent work perceptions—resilience; Path 3: career calling—career adaptability—future decent work perceptions—resilience.

## Data Availability

The data that support the outcomes of this study are accessible from the corresponding author. Restrictions apply to the availability of these data, which were utilized with license for this study. Data are available from the authors with the permission of Nanjing XiaoZhuang University.

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
