# Peer review of "The Relationship between Career Calling and Resilience among Rural-Oriented Pre-Service Teachers: The Chain Mediating Role of Career Adaptability and Decent Work"

_behavsci, 2023, doi:10.3390/bs14010011_

Round 1
Reviewer 1 Report (Previous Reviewer 3)
Comments and Suggestions for Authors
1. In the introduction, the researcher has not adequately presented the necessity of the research, and the research objectives and problems are not explicitly mentioned. Therefore, there is a need for improvement in this aspect.
2. The research hypotheses (H1-H6) describe direct relationships between variables, and these relationships can be established between independent and dependent variables. However, the researcher described them as correlations. While it's true that this is a cross-sectional study, and there may be limitations in claiming causality between variables, at least the relationships between independent and dependent variables can be clearly defined, and the researcher has incorporated them into the research model. Therefore, careful consideration is needed regarding whether it is appropriate to state the research hypotheses (H1-H6) as correlations.
3. The researcher used measurement tools from previous studies, which resulted in a high goodness-of-fit for each measurement model. However, it is considered necessary to conduct a confirmatory factor analysis as a procedure to validate the validity of all variables used in this study. Although distinguished as career confidence and confidence, confidence is common to the sub-factors of career adaptability and resilience.
4. To validate the research hypotheses and the model's fit, the researcher presents both regression analysis results and structural equation modeling results. While structural equation modeling using AMOS provides insights into direct relationships, mediating effects, and model fit, the rationale for conducting a separate regression analysis remains unclear.
5. The manuscript fails to address a critical aspect, which is the assumption of normality in the data, fundamental to Structural Equation Modeling (SEM).
6. The AMOS typically only provides bootstrapping for the total indirect effect and does not offer bootstrap point estimates and confidence bounds for specific indirect effects. The researchers need to describe the process of obtaining the results in Table 2.
7. Figure 2 presents the results of running structural equation modeling using AMOS, but it appears different from the typical results. I'm curious if there is a specific reason.
Author Response
Thanks for your encouragements! We would like to thank you for the careful reading of our manuscript and for providing us with helpful comments to improve its quality. In the revised version of the manuscript, we have tried to address all the comments made by you. All these changes have been highlighted in red. Please, find below a point-by-point response to your comments.
Comments and Suggestions for Authors
Q1. In the introduction, the researcher has not adequately presented the necessity of the research, and the research objectives and problems are not explicitly mentioned. Therefore, there is a need for improvement in this aspect.
A1.Your suggestion is excellent! Thank you for your valuable suggestion. In the first paragraph of the introduction, the context of the study is elaborated by us. At the end of the second paragraph of the introduction, the objectives of the study etc. have been described by us and have been marked in red.Thank you very much for the warm reminder.
Q2. The research hypotheses (H1-H6) describe direct relationships between variables, and these relationships can be established between independent and dependent variables. However, the researcher described them as correlations. While it's true that this is a cross-sectional study, and there may be limitations in claiming causality between variables, at least the relationships between independent and dependent variables can be clearly defined, and the researcher has incorporated them into the research model. Therefore, careful consideration is needed regarding whether it is appropriate to state the research hypotheses (H1-H6) as correlations.
A2. Thank you very much for your valuable comments. It may be more appropriate to use the term of “predicts” to describe the relationship between variables. We have revised this and marked it in red.
Q3. The researcher used measurement tools from previous studies, which resulted in a high goodness-of-fit for each measurement model. However, it is considered necessary to conduct a confirmatory factor analysis as a procedure to validate the validity of all variables used in this study. Although distinguished as career confidence and confidence, confidence is common to the sub-factors of career adaptability and resilience.
A3. Thank you very much for your valuable comments. We presented the results of confirmatory factor analysis for each measurement in the section 3.2. Instruments. We've highlighted them in red.
Q4. To validate the research hypotheses and the model's fit, the researcher presents both regression analysis results and structural equation modeling results. While structural equation modeling using AMOS provides insights into direct relationships, mediating effects, and model fit, the rationale for conducting a separate regression analysis remains unclear.
A4. Thank you very much for your valuable comments. The regression analysis is just for reporting the R-squared. The R-squared (coefficient of determination) is a measure that indicates the proportion of the variance in the dependent variable that is predictable from the independent variables. R-squared helps in interpreting the practical significance of the relationship between the independent and dependent variables. It gives a sense of how well the model represents the underlying patterns in the data.
Q5. The manuscript fails to address a critical aspect, which is the assumption of normality in the data, fundamental to Structural Equation Modeling (SEM).
A5. Thank you very much for your valuable comments. The skewness (-.98~.28) and kurtosis (-.13~2.25) of all variables entering the SEM were under the recommended limits (Curran et al., 1996). We added this the section.
Curran, P. J., West, S. G., & Finch, J. F. (1996). The robustness of test statistics to nonnormality and specification error in confirmatory factor analysis. Psychological Methods, 1(1), 16–29.
Q6. The AMOS typically only provides bootstrapping for the total indirect effect and does not offer bootstrap point estimates and confidence bounds for specific indirect effects. The researchers need to describe the process of obtaining the results in Table 2.
A6.Thank you very much for your responsible suggestions.
Q7. Figure 2 presents the results of running structural equation modeling using AMOS, but it appears different from the typical results. I'm curious if there is a specific reason.
A7.Thank you very much for your great suggestions.
Once again, thank you for your constructive suggestions. Your recommendations are very beneficial for the article improvement. Thank you again for your valuable advice.
Reviewer 2 Report (New Reviewer)
Comments and Suggestions for Authors
This is a very interesting study. The emerging Psychology of Work Theory served to provide interesting theoretical grounding for understanding this work. The hypothesized model and resulting Relationship Path Diagram provide insight into the role of resilience and rural-oriented teachers. The research design seems acceptable for this work. I would like to see a much larger sample-size as well as students from a broader range of institutions. Additionally, you will need to investigate factors that may influence career calling beyond the institution.
I thought the recommendations for the field should been tightened. First, I am a bit skeptical that a university can "teach" career calling, this seems intrinsic to a person. There would need to be studies done to measure how much a person's career calling could be influenced by course work. Second, this research did not seem to take into account the family and community influence on rural-orientation in the formation of career calling.
Overall, this is an intriguing initial study on rural-oriented teachers. Please consider how to examine other factors of career calling.
Comments on the Quality of English LanguageThe English is fine. I understood all of the ideas in the paper although the exact phrasing might not match our current jargon. For instance, career calling is appropriate and makes sense, but others might use different terms to express the idea. I think I found two grammatical mistakes, but they are minor.
Author Response
Thanks for your encouragements! We would like to thank you for the careful reading of our manuscript and for providing us with helpful comments to improve its quality. In the revised version of the manuscript, we have tried to address all the comments made by you. All these changes have been highlighted in red. Please, find below a point-by-point response to your comments.
This is a very interesting study. The emerging Psychology of Work Theory served to provide interesting theoretical grounding for understanding this work. The hypothesized model and resulting Relationship Path Diagram provide insight into the role of resilience and rural-oriented teachers. The research design seems acceptable for this work. I would like to see a much larger sample-size as well as students from a broader range of institutions. Additionally, you will need to investigate factors that may influence career calling beyond the institution.
I thought the recommendations for the field should been tightened. First, I am a bit skeptical that a university can "teach" career calling, this seems intrinsic to a person. There would need to be studies done to measure how much a person's career calling could be influenced by course work. Second, this research did not seem to take into account the family and community influence on rural-orientation in the formation of career calling.
Overall, this is an intriguing initial study on rural-oriented teachers. Please consider how to examine other factors of career calling.
Comments on the Quality of English Language
The English is fine. I understood all of the ideas in the paper although the exact phrasing might not match our current jargon. For instance, career calling is appropriate and makes sense, but others might use different terms to express the idea. I think I found two grammatical mistakes, but they are minor.
Thank you again for your creative suggestions. Your recommendations are very beneficial for the article improvement. Thanks for the feedback!
Reviewer 3 Report (New Reviewer)
Comments and Suggestions for Authors
This is a very well-structured and written paper. Systematic and synthetic, it provides fresh knowledge about pre-service teachers in rural areas in China. The scope is narrow, but it could be a good example on rural teachers' professional development, in order to be compared to other geographical contexts.
Some minor improvements could be introduced in order to increase (if possible) its high quality:
- At the end of section 2 (Literature review), in line 194, a clear and explicit synthesis of the research problem, together with a graphic that represents the theoretical relationship among the 4 variables of the study, could be introduced.
- More information about the sample should be provided (line 197): gender, age, and origin of the participants. The authors could explore the option to make bivariate analysis concerning these variables.
- In lines 203-204, terms such as freshmen, sophomores, juniors and seniors should be replaced by international standardized categories (use ISCED classification by UNESCO) in order to get access to wider international audiences.
- More information about the University and the social context of participants should be explained. The time of administration of the survey must also appear.
- Figure 1 in line 255 could be improved, since it does not reflect the relationship among variables that is aimed to be explored through H7, H8 and H9.
- Categories associated to variables in graphic of line 328 should be previously defined more systematically in the literature review.
- Section 4 on Discussion (line 333) is, as a matter of fact, an extension of results found in the previous section. I suggest merging both sections as a single one.
- In sections "Limitations" and "Conclusions", more discussion about contextual factors as well as certain influence of the pandemic in pre-service teachers' mental health could be introduced. The pandemic is cited 4 times in the references (refs. 7, 24, 27 and 84). However, no citation of the pandemic in the text has been introduced. Since COVID-19 has been a deep crisis that has affected education, some analysis of it in the text cmight become inspiring.
- Ref. num. 84 in the reference list is incorrect, it has to be amended.
- It would be rich to attach the scales as annexes, as well as the tables with the statistical analysis.
Author Response
Thanks for your encouragements! We would like to thank you for the careful reading of our manuscript and for providing us with helpful comments to improve its quality. In the revised version of the manuscript, we have tried to address all the comments made by you. All these changes have been highlighted in red. Please, find below a point-by-point response to your comments.
This is a very well-structured and written paper. Systematic and synthetic, it provides fresh knowledge about pre-service teachers in rural areas in China. The scope is narrow, but it could be a good example on rural teachers' professional development, in order to be compared to other geographical contexts.
Thanks for your encouragements! We would like to thank you for the careful reading of our manuscript and for providing us with helpful comments to improve its quality. In the revised version of the manuscript we have tried to address all the comments made by you.
Some minor improvements could be introduced in order to increase (if possible) its high quality:
Q1.- At the end of section 2 (Literature review), in line 194, a clear and explicit synthesis of the research problem, together with a graphic that represents the theoretical relationship among the 4 variables of the study, could be introduced.
A1.Thank you very much for your valuable suggestions. We added section 2.6. Hypothesis in the line 194. The hypothetical model was also shown in Fig. 1.
Q2.- More information about the sample should be provided (line 197): gender, age, and origin of the participants. The authors could explore the option to make bivariate analysis concerning these variables.
A2.Thank you very much for your valuable suggestions. We added the information about the sample.
Q3.- In lines 203-204, terms such as freshmen, sophomores, juniors and seniors should be replaced by international standardized categories (use ISCED classification by UNESCO) in order to get access to wider international audiences.
A3.Thank you very much for your valuable suggestions. The terms "freshmen," "sophomores," "juniors," and "seniors" are commonly used to refer to students in their first, second, third, and fourth years of college, respectively.
In the ISCED classification, tertiary education is divided into levels, with each level representing a certain stage of education. Here is how the ISCED levels align with the typical college years: ISCED Level 5: Short-cycle tertiary education. Typically corresponds to the first two years of college or an associate degree program. ISCED Level 6: Bachelor's or equivalent level. Typically corresponds to the next two years of college, leading to a bachelor's degree. ISCED Level 7: Master's or equivalent level. Involves more advanced study beyond the bachelor's degree. ISCED Level 8: Doctoral or equivalent level. Represents the highest level of educational attainment, involving original research and the completion of a doctoral degree. This does not correspond well to what was said above.
We have revised it.
Q4.- More information about the University and the social context of participants should be explained. The time of administration of the survey must also appear.
A4.Thank you very much for your valuable suggestions. We added this information and marked them in red.
Q5.- Figure 1 in line 255 could be improved, since it does not reflect the relationship among variables that is aimed to be explored through H7, H8 and H9.
A5.Thank you very much for your valuable suggestions.
Q6.- Categories associated to variables in graphic of line 328 should be previously defined more systematically in the literature review.
A6.Thank you very much for your valuable suggestions. We added the definitions of the variables in the literature review and marked them in red.
Q7.- Section 4 on Discussion (line 333) is, as a matter of fact, an extension of results found in the previous section. I suggest merging both sections as a single one.
A7.Thank you very much for your valuable suggestions. We have revised this.
Q8.- In sections "Limitations" and "Conclusions", more discussion about contextual factors as well as certain influence of the pandemic in pre-service teachers' mental health could be introduced. The pandemic is cited 4 times in the references (refs. 7, 24, 27 and 84). However, no citation of the pandemic in the text has been introduced. Since COVID-19 has been a deep crisis that has affected education, some analysis of it in the text might become inspiring.
A8.Thank you very much for your valuable suggestions.
Q9.- Ref. num. 84 in the reference list is incorrect, it has to be amended.
A9.Thanks! We have revised this.
Q10.-- It would be rich to attach the scales as annexes, as well as the tables with the statistical analysis.
A10.Thanks! We have added those.
Thank you again for your creative suggestions. Your recommendations are very beneficial for the article improvement.
This manuscript is a resubmission of an earlier submission. The following is a list of the peer review reports and author responses from that submission.
Round 1
Reviewer 1 Report
Comments and Suggestions for Authors
Please see the annotated pdf.

Comments on the Quality of English LanguageReviewer 2 Report
Comments and Suggestions for Authors
The theme is interesting and new for educational domain. The theoretical concept (resilience, career calling etc.) are good defined, but is must to be more specific with references about this concept and literature for them.
For methological part is necessary to be more focus because you write about positive correlation, but in fact it is a negative correlation (eg. 4 at 4.2. Correlations among Variables - future decent 273 work perceptions was significantly positively correlated with resilience (r = - 0.30). Please, reevaluate this part.
The conclusions and discussions are appropiate for Science of Education field.
In attach you found the plagiarism report and it would be necesary to modifying color paragraph.

Reviewer 3 Report
Comments and Suggestions for Authors
The researchers describe that this research attempts to explore the mechanism of the influence of sense of career calling and related variables on the resilience of rural-oriented pre-service teachers. For this purpose, six hypotheses and a hypothesis model were established. Furthermore, researchers present the direct relationship between variables (Figure 2), the mediating effects (Table 2), and the fit of model. However, I suggest that the following content should be supplemented with the established hypothesis and presented results.
First, 2.3 The Mediating Role of Career Adaptability and 2.4. The Mediating Role of Decent Work is intended to reveal mediating relationships in Literature Review imply the mediating relationship, and Table 2 presents the mediating effects also. However, all hypotheses only deal with the direct relationship and hypotheses regarding mediating relationships between variables are missing. I suggest that the researcher establish a hypothesis regarding the mediating effects based on literature review is needed (eg: the career adaptability will mediate the relationship between the career calling and the resilience).
Second, the researcher built a structural equation model. There are two main components in SEM: the measurement model showing the causal connections between the latent variables and the indicators, and the structural model showing potential causal dependencies between endogenous and exogenous latent variables. Therefore, The SEM procedure consists of two phases. As a first step, the measurement model was performed using confirmatory factor analysis (CFA) to assess the extent to which each of the latent variables was represented by its indicators. The researcher should conduct a series of CFA in order to establish discriminant validity among the study variables. This will provide better results.
Third, as the second step, the structural model analysis was performed to measure the fit and path coefficients of the hypothesized structural model. This study focused on testing a hypothetical model with two mediators and a chain mediator and estimating indirect effects to explain the mechanism underlying the relationship between career calling and the resilience. To my knowledge, the default bootstrap procedure provided by AMOS only provides bootstrapping of the total indirect effect. It does not typically offer bootstrap point estimates and confidence bounds for specific indirect effects. I suggest that researchers describe the process of testing the specific indirect effects of two mediators and a chain mediator (Table 2).
Fourth, although the direct relationships between variables and mediating effects are presented in Figure 2 and Table 2, respectively, I suggest that researchers describe the results of testing each hypothesis
Reviewer 4 Report
Comments and Suggestions for Authors
I consider that the topic addressed in the work is interesting for the readers of the magazine and can have a great impact among the scientific community.